# Socio-demographic and behavioral correlates of excess weight and its health consequences among older adults in India: Evidence from a cross-sectional study, 2017–18

Amiya Saha[1]*, T. Muhammad[1], Bittu Mandal[2], Mihir Adhikary[3], Papai Barman[1]

**1** Department of Family & Generations, International Institute for Population Sciences, Mumbai, India, **2** Indian Institute of Technology, School of Humanities and Social Sciences, Indore, India, **3** Department of Public health and Mortality Studies, International Institute for Population Sciences, Mumbai, India

* amiyasaha4444@gmail.com

## Abstract

### Background

Rapid population aging is expected to become one of the major demographic transitions in the twenty-first century due to the continued decline in fertility and rise in life expectancy. Such a rise in the aged population is associated with increasing non-communicable diseases. India has suffered from obesity epidemic, with morbid obesity affecting 5% of the population and continuing an upward trend in other developing countries. This study estimates the prevalence of excess weight among older adults in India, and examines the socio-demographic and behavioral factors and its health consequences.

### Methods

The study used data from the Longitudinal Ageing Study in India (LASI) wave 1 (2017–18). A total sample of 25,952 older adults ($\geq$ 60 years) was selected for the study. Descriptive statistics, bivariate Chi-Square test, and logistic regression models were applied to accomplish the study objectives. Body mass index (BMI) has been computed for the study according to the classification of the World Health Organization, and "excess weight" refers to a score of BMI $\geq$ 25.0 kg/m$^2$.

### Results

Overall, 23% of older adults ($\geq$ 60 years) were estimated with excess weight in India, which was higher among women irrespective of socioeconomic and health conditions. The higher levels of excess weight (*than the national average of* $\geq$22.7%) were observed among older adults in states like Haryana, Tamil Nadu, Telangana, Maharashtra, Gujarat, Manipur, Goa, Kerala, Karnataka, Himachal Pradesh, Punjab, Sikkim and some other states. After adjusting for selected covariates, the odds of excess weight were higher among females than males [OR: 2.21, 95% CI: 1.89, 2.60]. Similarly, the likelihood of excess weight was 2.18 times higher among older adults who were living in urban areas compared to their rural

**Data Availability Statement:** The data are available in the public domain and freely accessible from the Gateway to Global Aging Data (www.g2aging.org).

The data are also available in the International Institute for Population Sciences (IIPS) data repository of Longitudinal Aging Study in India, upon request to IIPS, Mumbai, https://www.iipsindia.ac.in/content/LASI-data.

**Funding:** The authors received no specific funding for this work.

**Competing interests:** The authors have declared that no competing interests exist.

counterparts [OR: 2.18; 95% CI: 1.90, 2.49]. Higher level of education is significantly positively correlated with excess weight. Similarly, higher household wealth index was significantly positively correlated with excess weight [OR: 1.98, CI: 1.62, 2.41]. Having hypertension, diabetes and heart diseases were associated with excess weight among older adults. Regional variations were also observed in the prevalence of excess weight among older adults.

## Conclusion

The findings suggest that introducing measures that help to reduce the risk of non-communicable diseases, and campaigns to encourage physical activity, and community awareness may help reduce the high burden of excess weight and obesity among older Indians. The findings are important for identifying the at-risk sub-populations and for the better functioning of any public health programme and suitable intervention techniques to lower the prevalence and risk factors for excess weight in later life.

## Background

Rapid population aging is expected as a result of the major demographic transitions in the twenty-first century due to the continued decline in fertility and rise in life expectancy [1]. By 2100, there will be 3.1 billion people over the age of sixty, which is a threefold increase from 2017 [2]. Asian countries have a higher rate of population aging and a higher absolute number of older persons, despite the relative share of the older population being higher in Western countries [3]. About 104 million people in India are 60 or older, constituting 8.6% of the total population, and by 2050, the percentage is expected to rise to 20% of the population [4].

An increase in noncommunicable diseases is associated with population aging and increased life expectancies [5]. A significant rise in morbidities is brought on by obesity, which is a leading lifestyle disease worldwide [6] and has recently grown to be a major global public health concern [7,8]. It is considered the main factor contributing to the onset and severity of noncommunicable diseases [9]; obesity also raises mortality risks and affects the quality-of-life years [10]. For the past three decades, advanced regions like the USA and Europe have suffered from a severe obesity problem [11] and it was also reported that the majority of the world's population lives in countries where issues associated with obesity affect more individuals than those who are underweight [12]. The World Health Organization (WHO) reports the prevalence of obesity has tripled globally between 1975 and 2016 [13]. The prevalence of obesity, traditionally thought to be a concern in developed countries, is becoming a major public health challenge in low- and middle-income countries [14]. In 2019, 5.02 million people died prematurely owing to obesity, nearly six times as many as from HIV/AIDS, according to the Global Burden of Disease (GBD) study [15]. Over 8% of all deaths globally in 2019 were related to obesity; the figure was merely 4% in 1990 [16].

India also suffered from obesity epidemic, with morbid obesity affecting 5% of the population in the twenty-first century and is continuing an upward trend seen in other developing countries [17]. It has also seen a greater increase in the prevalence of obesity than the global average. An Indian study of older adults found that the prevalence of excess weight ($\geq$25.0 kg/m$^2$) was 14% in 2007 [18]. These could be different from younger population groups due to changes in body composition, height, food intake, and energy expenditure that occur as people age [19,20] and they can be prevented through behavioral and lifestyle changes [21]. Communities and environments supporting healthy lifestyle choices are essential in people's

perception [21]. In addition to having more body fat, older persons also have altered distributions of that fat; similarly, aging is associated with loss of height and mass [22]. The average calorie intake and hunger level in older adults are often lower. Furthermore, as people age, their level of physical activity declines [22]. Due to their frailty, sickness, and impending death, older adults usually lose weight over time [22]. Studies from developed countries reveal that obesity may negatively affect morbidity more than mortality in later life [23,24]. Previous studies have also found associations between obesity, depression [25] and diminished quality of life [26] among older adults.

Although more information is available on the physical, social, and economic factors that are associated with higher body mass index (BMI) scores in younger people [24], there is a dearth of knowledge on how patterns of obesity differ across different segments of the older population [27]. Himes [28] found that older women are more likely to be overweight and obese than older males, according to data from the Assets and Health Dynamics of the Oldest Old Survey and the Longitudinal Study of Aging. As documented, female sex, better socioeconomic status, and living in an urban area are important socio-demographic factors associated with higher BMI levels [29], and the behavioral factors include physical inactivity and smoking whereas, health consequences are poor self-rated health and non-communicable diseases such as hypertension, heart disease and diabetes [29].

Understanding the prevalence of excess weight and its risk factors and health consequences in older adults is necessary to frame targeted policies and programs to reduce the morbidity and mortality related to excess weight. Therefore, this study aimed to a) assess the prevalence of excess weight among community-dwelling older adults in India and its states; and to b) determine the socio-demographic and behavioral factors of excess weight including gender, age, education, marital status, smoking, health status and place of residence, and health consequences of excess weight among older adults.

## Methods

### Data source

The Longitudinal Aging Study in India (LASI) wave 1 (2017–18), a national and state-representative survey of aging and health, provided the data for the current study. In its initial round, the LASI surveyed 72,250 samples of adults 45 and over throughout all 35 Indian states and union territories (UT) [30]. The major goal of the LASI survey was to offer longitudinal, valid, and reliable information on the socioeconomic and health status of the older Indian population. The LASI employed a multistage stratified area probability cluster sampling design to determine the final units of observation. LASI employed a three-stage sample design in rural areas, while in urban areas, it employed a four-stage sample design. Primary Sampling Units (PSUs) were chosen in each state and UT in the first stage. In the second stage, villages in rural regions and wards in urban areas were chosen in the selected PSUs. In the third round, households in selected villages were selected in rural areas. Urban sampling, however, required an additional step. One Census Enumeration Block (CEB) was specifically chosen at random in each urban region during the third stage. From this CEB, households were chosen as a fourth stage. The survey report included the complete methodology, including all details on the survey's design and data collection [30].

### Study population

The current study used secondary data, specifically LASI Wave 1, which has a total sample of 73,396 adults aged 45 years and older and their spouses, regardless of age, with no missing values in age reporting. The older adults were reached out at their houses during the face-to-face

interviews [30]. In this study, the participants were older adults, 60 years of age and above, who provided detailed information on their biometric measurements. After removing older adults less than 60 years (n = 37,924), those who had incomplete information on BMI (n = 532), and those who also provided incomplete information on other factors associated with excess weight (n = 2,451), the number of participants in this study included 25,952 older adults. Fig 1 shows the inclusion and exclusion criteria for the study sample.

### Ethics statement

The study is based on publicly available data (https://g2aging.org/), and survey organizations that carried out the field survey for the purpose of data collection also obtained the respondent's prior agreement. Ethics approval was obtained from the Central Ethics Committee on Human Research (CECHR) under the Indian Council of Medical Research (ICMR) and the Institutional Review Boards of collaborating organizations including the International Institute for Population Sciences (IIPS), Mumbai and the Ministry of Health and Family Welfare,

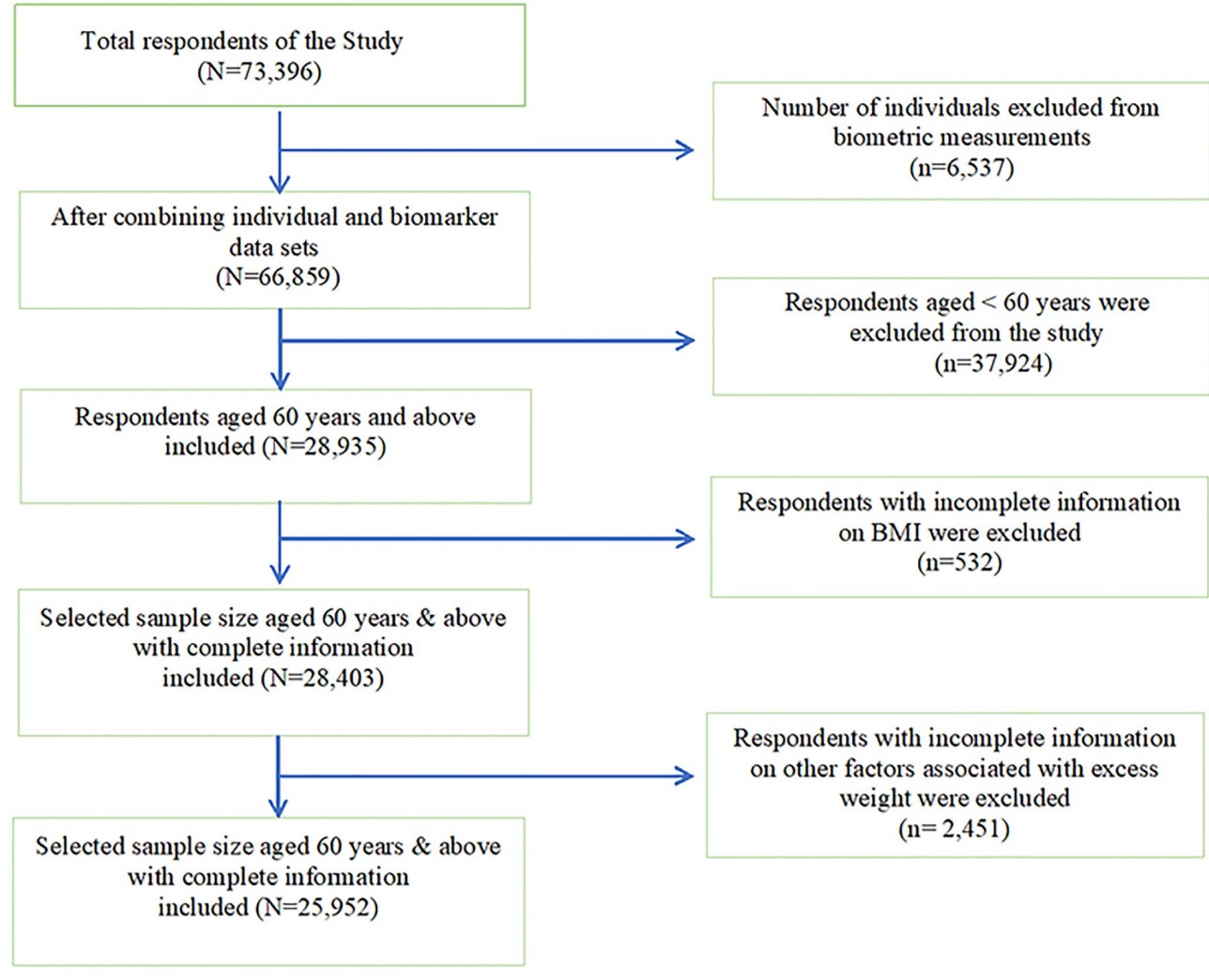

**Fig 1. Selection criteria of the sample study.**

Government of India. All processes related to the survey were carried out in accordance with the relevant guidelines and regulations of ICMR. With support from the Ministry of Health and Family Welfare (MoHFW), the International Institute for Population Sciences (IIPS), the United Nations Population Fund (UNFPA), and other organisations, the LASI was completed.

## Variable description

**Outcome variable.** The outcome of interest i.e., body mass index (BMI) was measured based on height and weight of older adults. "Height and weight of adults were measured using the Seca 803 digital scale" [30]. It was categorized according to the classification of the World Health Organization: underweight ($<18.5$ kg/m$^2$), normal weight (18.5–24.9 kg/m$^2$), over-weight (25.0–29.9kg/m$^2$), obesity ($\geq$30.0 kg/m$^2$) [31]. It was further coded as 0 "no-excess weight" if the older adults had a score of BMI $\leq$ 24.9 kg/m$^2$ and "excess weight" as 1 if the older adults had a score of BMI $\geq$ 25.0 kg/m$^2$ [32].

**Other measures.** The study included three sets of variables: (1) socio-demographic; (2) behavioral; and (3) health-related variables. A detailed description of the predictor variables appears in Table 1.

## Statistical analysis

Descriptive statistics and bivariate analysis were used to evaluate the prevalence of excess weight by socioeconomic status and health and behavioral factors. The significance level of the bivariate associations were determined using Chi-Squared tests. In addition, binary logistic regression analysis was used to examine the associations between different socio-demographic and behavioral factors and health-related consequences of excess weight among older adults.

While examining the possible determinants of excess weight, model 1 provides the univariate association of excess weight with the socioeconomic and behavioral characteristics of older adults. Model 2 (full model) was controlled for all the selected covariates in this study and provides the adjusted associations of excess weight with the socio-demographic and behavioral characteristics of older adults. An additional table (**Table 5**) is provided to report the health consequences of excess weight among older adults. All the statistical analysis was performed using STATA version 16.0 (Stata Corp, LP, college station, Texas), and ArcGIS 10.8 software for the state-level mapping.

## Results

### Socioeconomic and demographic profile of older adults

Table 2 presents the socioeconomic and demographic profile of older adults. Around 23% of the older adults had excess weight. More than 55% of participants were women. Almost two third of the older adults had no education, and nearly 38% of older adults were not in a marital union. Around 22% of the older adults belonged to the lowest stratum of household wealth. Around 68% of the older adults never did physical activity. Around 13% of the older adults were current smokers, nearly 21% of the older adults consumed smokeless tobacco, while 3% of the older adults consumed both. Moreover, 6%, 13% and around 1% older adults were frequent, infrequent, and heavy alcohol drinker, respectively. More than half of the older adults reported being diagnosed with hypertension, while 14% and 5% older adults had diabetes and heart diseases, respectively. More than 23% of the older adults reported poor self-rated health at the time of survey.

**Table 1. Description of the explanatory factors included in the study, Longitudinal Aging Study (LASI) Wave 1, India 2017–18.**

| Socio-demographic | | |
|---|---|---|
| | **Categories** | **Description of the category** |
| **Sex** | Male | Sex of the respondent was available in male-female categories. |
| | Female | |
| **Place of residence** | Rural | Place of residence (rural/urban) was determined according to the administrative division of India followed in Census of India, 2011. Households in urban areas included those in towns, wards and Census Enumeration Blocks whereas, households in rural areas include those in villages (size varies from 0–10,000 population [30]. |
| | Urban | |
| **Religion** | Hindu | Religion was categorized into Hindu, Muslim, Christian, and others [33]. |
| | Muslim | |
| | Christian | |
| | Others | |
| **Caste** | Scheduled castes (SC) | Caste was coded as Scheduled castes (SC), Scheduled Tribes (ST), Other Backward Class (OBC) and others. SC and ST are India's most economically and socially disadvantaged groups. According to the Hindu caste system, the ST contains a segment of the population that is socially isolated and has a low economic position. People who were "educationally, economically, and socially backward" are classified as OBC. In the old caste order, the OBC is seen as being at the bottom yet somewhat above the most disadvantaged communities. The "other" caste category is identified as those having higher social standings [34,35]. |
| | Scheduled Tribes (ST) | |
| | Other backward classes (OBC) | |
| | Others | |
| **Education** | No education | There were four categories for educational status: no education, up to primary, up to secondary and secondary and above. |
| | Up to primary | |
| | Up to secondary | |
| | Secondary and above | |
| **Marital status** | Currently in union | Our study has categorized marital status into binary, "Currently in union" those who responded as "currently married" and all other categories as "Currently not in union" including those who responded as widowed, never married, separated, divorced, and deserted [36]. |
| | Currently not in union | |
| **MPCE quintile** | Lowest | Using information on household consumption data, the monthly per capita consumption expenditure (MPCE) quintile has been assessed. The sample households were surveyed using sets of 11 and 29 questions on spending on food and non-food items, respectively. Food expenditures were collected during the seven-day reference period, whilst non-food expenditures were collected over 30-day and 365-day reference periods. Using 30-day reference period, expenses for both food and non-food items were standardised. The MPCE is calculated and used as a summary indicator of consumption. The variable was further divided into five quintiles, i.e., from lowest to highest [30]. |
| | Lower | |
| | Middle | |
| | Higher | |
| | Highest | |
| **Regions** | North | The regions were coded as North, West, Northeast, East, Central, and South. |
| | West | |
| | Northeast | |
| | East | |
| | Central | |
| | South | |
| Behavioral | | |

(*Continued*)

**Table 1.** (Continued)

| Socio-demographic | | |
| --- | --- | --- |
| | **Categories** | **Description of the category** |
| **Physical activity** | Frequent | There were three categories for physical activity categories: regular (every day), rare (more than once a week, once a week, once to three times a month), and never. Physical activity was assessed using the following question: "How often do you take part in sports or vigorous activities, such as running or jogging, swimming, going to a health centre or gym, cycling, or digging with a spade or shovel, heavy lifting, chopping, farm work, fast bicycling, cycling with loads?" [30]. |
| | Rare | |
| | Never | |
| **Consumption of tobacco** | Never consumed tobacco | Consumption of tobacco was categorised by asking three questions to the older adults during survey; (i) Have you ever smoked tobacco (cigarette, bidi, cigar, hookah, cheroot) or used smokeless tobacco (such as chewing tobacco, gutka, pan masala, etc.)?" Those who responded no was coded as "never consumed tobacco." (ii) "What type of tobacco product have you used or consumed?" Those who responded Smokeless tobacco (such as chewing tobacco, gutka, pan masala, etc.) was coded as "Currently consumed smokeless tobacco" and both Smoke and smokeless tobacco was coded as "Consumed both smoking and smokeless tobacco." (iii) "Do you currently smoke any tobacco products (cigarettes, bidis, cigars, hookah, cheroot, etc.)?" Those who responded yes was coded as "currently smoking" [30]. |
| | Currently smoking | |
| | Currently consumed smokeless tobacco | |
| | Consumed both smoking and smokeless tobacco | |
| **Consumption of alcohol** | Never consumed alcohol | Similarly, consumption of alcohol was categorised by asking three questions to the older adults during survey; (i) Have you ever consumed any alcoholic beverages such as beer, wine, liquor, country liquor etc.?" (ii) "In the past three months, on an average, how frequently [on how many days], have you had at least one alcoholic drink? (For example, beer, wine, or any drink, such as country liquor, containing alcohol.?." (iii) "In the last 3 months, how frequently on average, have you had at least 5 or more drinks on one occasion?" Those who responded no was coded as "never consumed alcohol." Consumed none, less than once a month in past three months was coded as "frequently consumed but not a heavy drinker"; those who drank one to four times a week, one to four times a day, or five or more times a day but did not drink more than five drinks at once in the previous 30 days was coded as "Infrequently consumed but not a heavy drinker" and those who, at least once during the previous 30 days, consumed five or more alcoholic beverages was coded as "heavy drinker" [30]. |
| | Frequently consumed but not a heavy drinker | |
| | Infrequently consumed but not a heavy drinker | |
| | Heavy drinker | |
| **Health-related** | | |
| **Hypertension** | Yes | Hypertension was defined based on measured and self-reported information, as i. those who had a mean systolic blood pressure $\geq$140 mmHG and/ or mean diastolic blood pressure $\geq$90 mmHG (based on the last two averaged of three readings) [37]; ii. ever diagnosed with hypertension by any health professionals (yes and no) [30] and iii. taking any medications or under any diet restrictions (yes and no) [30]. If the older adults fall into any of these three, then they were considered as "having hypertension" and others were categorised as "no." |
| | No | |
| **Diabetes** | Yes | Diabetes was sourced from the question asked during the survey "Has any health professional ever told you that you have diabetes?." Older adults were asked to respond yes and no [30]. Those who reported "yes" were classified as diabetic and otherwise no. |
| | No | |

(*Continued*)

**Table 1.** (Continued)

| Socio-demographic | | |
| --- | --- | --- |
| | Categories | Description of the category |
| Heart disease | Yes | Older adults were asked "Has any health professional ever told you that you have heart disease such as coronary heart disease?." Older adults were asked to respond yes and no [30]. Those who reported "yes" were classified as having heart disease and otherwise no. |
| | No | |
| Stroke | Yes | Similarly, stroke was also sourced by asking the question during survey "Has any health professional ever told you that you have Stroke?." Further, older adults were asked to respond yes and no [30]. Those who reported "yes" were classified as having diagnosed with stroke and otherwise no. |
| | No | |
| Self-rated health (SRH) | Good | Health status was assessed by self-reported health (SRH) [30]. In the survey, individuals were asked to rate their health between 1 to 5, where 1 denoted very good, and 5 denoted very bad. Thus 1 to 3 score (very good, good and fair) was coded as good, and 4 to 5 (poor and very poor) was coded as poor [36]. |
| | poor | |

## Prevalence of excess weight among older adults

Table 3 shows the prevalence of excess weight among older adults based on socioeconomic, behavioral and health characteristics. The prevalence of excess weight was higher among female (27%) than in male (18%) older adults in India. Older adults living in urban areas had a higher prevalence of excess weight than rural areas (40% vs. 16%). Prevalence of excess weight was high among other religions (32%) and other castes (29%), respectively. A higher percentage of older adults who were highly educated (42%) and were currently in a marital union (24%) had excess weight. The prevalence of excess weight was high among older adults belonging to the highest quintile (33%). Surprisingly, 28% and 25% of the older adults who never consumed any tobacco or alcohol had excess weight. A higher percentage of older adults with hypertension (30%), diabetes (46%), and heart diseases (41%) had excess weight. Additionally, the prevalence of excess weight was higher among the older adults who had a stroke (26) and never did physical activity (24%). Again, a slightly higher percentage of older adults who reported good SRH had excess weight (24%) compared to those who reported poor SRH (20%). The prevalence of excess weight was the highest in the south region (34%), followed by the west (29%) and north region (27%).

## State-wise prevalence of excess weight among older adults in India

Fig 2 illustrates the spatial distribution of excess weight prevalence across different states in India. States with a notable prevalence of excess weight higher than the national average of ≥22.7% include Haryana, Tamil Nadu, Telangana, Maharashtra, Jammu and Kashmir, Gujarat, Manipur, Goa, Andhra Pradesh, Kerala, Karnataka, Himachal Pradesh, Punjab, and Sikkim, particularly among the older adult population. Among older adults, a percentage ranging from 15% to below 22.7% exhibited excess weight in states like Madhya Pradesh, Odisha, Mizoram, Arunachal Pradesh, Uttarakhand, and Rajasthan. Conversely, states such as Meghalaya, Assam, Tripura, Nagaland, West Bengal, Chhattisgarh, Bihar, Jharkhand, and Uttar Pradesh displayed a comparatively lower prevalence of excess weight, falling below the 15% threshold among the older adult population.

**Table 2. Characteristics of the study sample of older adults (60 years and above) in India 2017–18.**

| Background | Sample (N) | Percentage (%) |
|---|---|---|
| **Excess weight** | | |
| No | 20,056 | 77.3 |
| **Yes** | 5,896 | 22.7 |
| **Sex** | | |
| Male | 11,653 | 44.9 |
| Female | 14,299 | 55.1 |
| **Place of residence** | | |
| Rural | 18,473 | 71.2 |
| Urban | 7,478 | 28.8 |
| **Religion** | | |
| Hindu | 21,494 | 82.8 |
| Muslim | 2,863 | 11.0 |
| Christian | 642 | 2.5 |
| Others | 952 | 3.7 |
| **Caste** | | |
| Schedule caste | 4,957 | 19.1 |
| Schedule tribe | 1,916 | 7.4 |
| Other backward class | 11,849 | 45.7 |
| Others | 7,229 | 27.9 |
| **Education** | | |
| No education | 17,774 | 68.5 |
| Up to primary | 2,877 | 11.1 |
| Up to secondary | 1,745 | 6.7 |
| Secondary & above | 3,556 | 13.7 |
| **Marital status** | | |
| Currently in union | 16,075 | 61.9 |
| Currently not in union | 9,877 | 38.1 |
| **MPCE quintile** | | |
| Lowest | 5,599 | 21.6 |
| Lower | 5,616 | 21.6 |
| Middle | 5,421 | 20.9 |
| Higher | 5,061 | 19.5 |
| Highest | 4,255 | 16.4 |
| **Physical activity** | | |
| Frequent | 6,162 | 23.7 |
| Rare | 2,064 | 8.0 |
| Never | 17,726 | 68.3 |
| **Tobacco consumption** | | |
| Never consumed tobacco | 16,171 | 62.3 |
| Currently smoking | 3,396 | 13.1 |
| Currently consumed smokeless tobacco | 5,573 | 21.5 |
| Consumed both smoking and smokeless tobacco | 813 | 3.1 |
| **Alcohol consumption** | | |
| Never consumed alcohol | 20,847 | 80.3 |
| Frequently consumed but not a heavy drinker | 1,593 | 6.1 |
| Infrequently consumed but not a heavy drinker | 3,329 | 12.8 |
| Heavy drinker | 183 | 0.7 |

(*Continued*)

**Table 2.** (Continued)

| Background | Sample (N) | Percentage (%) |
|---|---|---|
| **Hypertension** | | |
| No | 11,496 | 44.3 |
| Yes | 14,456 | 55.7 |
| **Diabetes** | | |
| No | 22,226 | 85.6 |
| Yes | 3,726 | 14.4 |
| **Heart disease** | | |
| No | 24,656 | 95.0 |
| Yes | 1,296 | 5.0 |
| **Stroke** | | |
| No | 25,360 | 97.7 |
| Yes | 592 | 2.3 |
| **SRH** | | |
| Good | 19,914 | 76.7 |
| Poor | 6,038 | 23.3 |
| **Region** | | |
| North | 3,326 | 12.8 |
| West | 4,345 | 16.7 |
| Northeast | 722 | 2.8 |
| East | 6,262 | 24.1 |
| Central | 5,604 | 21.6 |
| South | 5,694 | 21.9 |

## Factors associated with excess weight among older adults in India

Table 4 depicts the results obtained from the logistic regression analysis of the socio-demographic and behavioral factors associated with excess weight among older adults in India. Model-1 presents unadjusted estimates whereas, model 2 provides the adjusted estimates.

After adjusting for the selected covariates, the odds of excess weight were significantly higher among older females than males [OR: 2.21, 95% CI: 1.89, 2.60]. Similarly, the likelihood of excess weight was 2.18 times higher among the older adults who were living in urban areas with reference to their rural counterparts [OR: 2.18; 95% CI: 1.90, 2.49]. Older adults who belonged to Hindu and Muslim religious affiliations had a 0.73 [OR: 0.73, 95% CI: 0.59, 0.89] and 0.81 times [OR: 0.81, 95% CI: 0.62, 1.05] lower likelihood of excess weight compared to older adults of other religions. Compared to the other castes, older adults who belonged to the scheduled castes OR: 0.73, 95% CI: 0.60, 0.87], scheduled tribes [OR: 0.39, 95% CI: 0.31, 0.50], and other backward class [OR: 0.84, 95% CI: 0.73, 0.97] had a significantly lower likelihood of excess weight. The increasing level of education had a significant positive association with the likelihood of excess weight. Similarly, Older adults belonged to the highest stratum [OR: 1.98, 95% CI: 1.62, 2.41], higher stratum [OR: 1.84, 95% CI: 1.50, 2.26], middle stratum [OR: 1.48, 95% CI: 1.23, 1.78], and lower stratum [OR: 1.26, 95% CI: 1.05, 1.52] of household wealth index had a significantly higher likelihood of excess weight than older adults of lowest stratum. Also, people who consumed both smoking and smokeless tobacco were less likely to have excess weight than people who never consumed tobacco [OR: 0.64, 95% CI: 0.46, 0.87]. Compared to the northern region, older adults living in the North-eastern [OR: 0.55, 95% CI: 0.45, 0.68], eastern [OR: 0.53, 95% CI: 0.45, 0.62] and central region [OR: 0.56, 95% CI: 0.46, 0.67]

**Table 3. Prevalence of excess weight among older adults (aged 60 years and above).**

| Socio-demographic characteristics | Prevalence (%) | 95% CI | p-value |
|---|---|---|---|
| **Sex** | | | < 0.001 |
| Male | 17.6 | 16.3, 18.9 | |
| Female | 26.9 | 24.8, 29.2 | |
| **Place of residence** | | | < 0.001 |
| Rural | 15.7 | 14.9, 16.6 | |
| Urban | 40.0 | 36.5, 43.7 | |
| **Religion** | | | < 0.001 |
| Hindu | 22.4 | 20.8, 24.1 | |
| Muslim | 21.9 | 19.6, 24.5 | |
| Christian | 23.0 | 19.3, 27.1 | |
| Others | 31.7 | 28.1, 35.6 | |
| **Caste** | | | < 0.001 |
| Schedule caste | 15.6 | 13.7, 17.7 | |
| Schedule tribe | 8.26 | 6.9, 9.9 | |
| Other backward class | 24.0 | 21.4, 26.8 | |
| Others | 29.4 | 27.8, 31.1 | |
| **Education** | | | < 0.001 |
| No education | 17.1 | 16.1, 18.1 | |
| Up to primary | 26.3 | 23.5, 29.2 | |
| Up to secondary | 35.5 | 26.2, 46.0 | |
| Secondary & above | 41.7 | 36.8, 46.7 | |
| **Marital status** | | | 0.001 |
| Currently in union | 23.2 | 22.0, 24.4 | |
| Currently not in union | 22.0 | 19.0, 25.2 | |
| **MPCE quintile** | | | < 0.001 |
| Lowest | 14.6 | 13.2, 16.2 | |
| Lower | 18.4 | 16.8, 20.2 | |
| Middle | 22.6 | 20.2, 25.2 | |
| Higher | 28.2 | 24.2, 32.5 | |
| Highest | 32.8 | 28.5, 37.4 | |
| **Physical activity** | | | < 0.001 |
| Frequent | 20.5 | 17.9, 23.3 | |
| Rare | 18.1 | 15.8, 20.7 | |
| Never | 24.0 | 22.3, 25.9 | |
| **Tobacco consumption** | | | < 0.001 |
| Never consumed tobacco | 28.2 | 26.2, 30.3 | |
| Currently smoking | 9.7 | 8.3, 11.4 | |
| Currently consumed smokeless tobacco | 16.5 | 14.8, 18.3 | |
| Consumed both smoking and smokeless tobacco | 10.9 | 8.5, 13.7 | |
| **Alcohol consumption** | | | < 0.001 |
| Never consumed alcohol | 25.3 | 23.7, 27.0 | |
| Frequently consumed but not a heavy drinker | 17.5 | 15.1, 20.3 | |
| Infrequently consumed but not a heavy drinker | 9.8 | 8.4, 11.5 | |
| Heavy drinker | 6.7 | 3.9, 11.4 | |
| **Hypertension** | | | < 0.001 |
| **No** | 13.1 | 11.9, 14.3 | |
| **Yes** | 30.4 | 28.3, 32.6 | |

(*Continued*)

**Table 3.** (Continued)

| Socio-demographic characteristics | Prevalence (%) | 95% CI | p-value |
|---|---|---|---|
| **Diabetes** | | | < 0.001 |
| **No** | 18.8 | 17.8, 19.7 | |
| **Yes** | 46.4 | 40.7, 52.2 | |
| **Heart disease** | | | < 0.001 |
| **No** | 21.7 | 20.5, 23.1 | |
| **Yes** | 41.4 | 30.2, 53.5 | |
| **Stroke** | | | 0.002 |
| **No** | 22.6 | 21.2, 24.1 | |
| **Yes** | 26.3 | 21.5, 31.6 | |
| **SRH** | | | 0.034 |
| **Good** | 23.5 | 21.8, 25.3 | |
| **Poor** | 20.2 | 18.7, 21.6 | |
| **Region** | | | < 0.001 |
| North | 27.4 | 25.8, 29.1 | |
| West | 28.9 | 26.6, 31.3 | |
| Northeast | 14.4 | 12.6, 16.5 | |
| East | 14.2 | 12.9, 15.6 | |
| Central | 14.8 | 13.2, 16.5 | |
| South | 33.5 | 28.8, 38.6 | |
| **India (Total)** | 22.7 | | |

**Note**. p-value based on Pearson Chi-square ($\chi^2$) tests.

had lower odds of excess weight, but interestingly older adults from southern regions were 1.24 times higher odds [OR: 1.24, 95% CI: 1.04, 1.48] of having excess weight than the northern counterparts.

The odds ratios of the logistic regression models (Table 5) suggest that older adults with excess weight were 2.19 times [OR: 2.19, 95% CI: 1.92, 2.50], 2.14 times [OR: 2.14, 95% CI: 1.83, 2.50] and 1.63 times [OR: 1.63, 95% CI: 1.26, 2.11] significantly more likely to be hypertensive, diabetic and have heart disease. Similarly, older adults with excess weight were 0.85 times [OR: 0.85, 95% CI: 0.74, 0.98] significantly less likely to report good SRH in this study.

## Discussion

This study is an attempt to assess the prevalence of excess weight and to examine the socio-demographic and behavioral factors associated with excess weight and its health consequences among older Indian adults. While India ranked 107th among 121 countries on the 2022 Global Hunger Index (GHI), the current findings indicated that one in every four Indians aged 60 years and above had excess weight. Such a higher prevalence of excess weight among older adults is intriguing and motivates us to dig more into this significant public health problem of excess weight. Further, we observed that the percentage of women with excess weight is significantly higher than that of males [38–40]. In developing countries, women tend to be less active than men, which can contribute to their increased risk of being overweight and obese; however, in high-income countries, females are not disadvantaged when it comes to physical inactivity and healthy food habits; as resources and opportunities for men and women have become increasingly alike over the years [41].

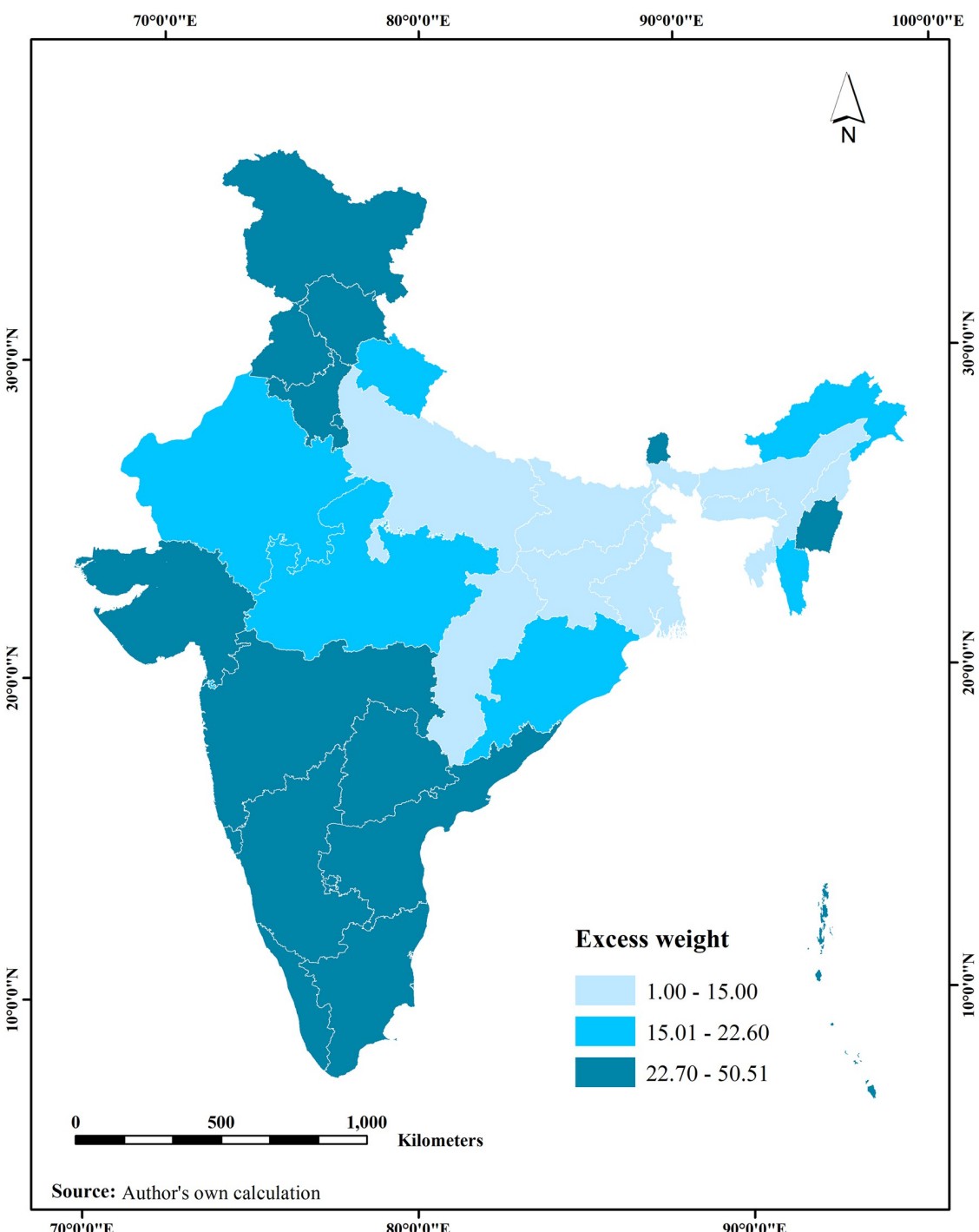

**Fig 2. Prevalence of excess weight among older adults in India by its states using LASI Wave 1 data.**

In India, the prevalence of excess weight is significantly related to their social and economic standing. Populations from a higher caste, households with the highest MPCE quintile, higher level of education, and residing in urban areas have a greater prevalence of excess weight than that belonging to lower castes, lower economic stratum, rural areas and are less educated

**Table 4. Socio-demographic and behavioral factors associated with excess weight among older adults (60 years and above) in India using logistic regression models, LASI, 2017–18.**

| Variables | b | Model—1 | p- value | b | Model—2 | p- value |
|---|---|---|---|---|---|---|
| | | Crude OR (95% CI) | | | Adjusted OR (95% CI) | |
| **Sex** | | | | | | |
| Male | ref | | | ref | | |
| Female | 0.54 | 1.72 [1.49, 1.99] | < 0.001 | 0.79 | 2.21 [1.89, 2.60] | < 0.001 |
| **Place of residence** | | | | | | |
| Rural | ref | | | Ref | | |
| Urban | 1.27 | 3.58 [3.04, 4.21] | < 0.001 | 0.78 | 2.18 [1.90, 2.49] | < 0.001 |
| **Religion** | | | | | | |
| Hindu | -0.47 | 0.62 [0.51, 0.76] | < 0.001 | -0.32 | 0.73 [0.59, 0.89] | < 0.05 |
| Muslim | -0.5 | 0.60 [0.48, 0.76] | < 0.001 | -0.21 | 0.81 [0.62, 1.05] | 0.117 |
| Christian | -0.44 | 0.64 [0.49, 0.85] | < 0.005 | -0.45 | 0.63 [0.44, 0.91]] | < 0.05 |
| Others | ref | | | Ref | | |
| **Caste** | | | | | | |
| Schedule caste | -0.82 | 0.44 [0.37, 0.52] | < 0.001 | -0.32 | 0.73 [0.60, 0.87] | < 0.005 |
| Schedule tribe | -1.53 | 0.22 [0.17, 0.27] | < 0.001 | -0.94 | 0.39 [0.31, 0.50] | < 0.001 |
| Other backward class | -0.28 | 0.76 [0.64, 0.89] | 0.001 | -0.17 | 0.84 [0.73, 0.97] | < 0.05 |
| Others | ref | | | ref | | |
| **Education** | | | | | | |
| No education | ref | | | ref | | |
| Up to primary | 0.55 | 1.73 [1.47, 2.04] | < 0.001 | 0.43 | 1.54 [1.29, 1.83] | < 0.001 |
| Up to secondary | 0.98 | 2.66 [1.71, 4.16] | < 0.001 | 0.94 | 2.57 [1.86, 3.55] | < 0.001 |
| Secondary & above | 1.24 | 3.46 [2.79, 4.30] | < 0.001 | 0.97 | 2.63 [2.12, 3.27] | < 0.001 |
| **Marital status** | | | | | | |
| Currently in union | ref | | | ref | | |
| Currently not in union | -0.07 | 0.93 [0.77, 1.13] | 0.471 | -0.37 | 0.69 [0.61, 0.79] | < 0.001 |
| **MPCE quintile** | | | | | | |
| Lowest | ref | | | ref | | |
| Lower | 0.28 | 1.32 [1.12,1.56] | 0.001 | 0.23 | 1.26 [1.05, 1.52] | < 0.05 |
| Middle | 0.53 | 1.71 [1.41, 2.06] | < 0.001 | 0.39 | 1.48 [1.23, 1.78] | < 0.001 |
| Higher | 0.83 | 2.29 [1.80, 2.92] | < 0.001 | 0.61 | 1.84 [1.50, 2.26] | < 0.001 |
| Highest | 1.05 | 2.86 [2.25, 3.62] | < 0.001 | 0.68 | 1.98 [1.62, 2.41] | < 0.001 |
| **Physical activity** | | | | | | |
| Frequent | ref | | | ref | | |
| Rare | -0.15 | 0.86 [0.68, 1.09] | 0.207 | -0.74 | 0.92 [0.70, 1.20] | 0.522 |
| Never | 0.21 | 1.23 [1.02, 1.49] | < 0.005 | 0.04 | 1.04 [0.89,1.22] | 0.62 |
| **Tobacco consumption** | | | | | | |
| Never consumed tobacco | ref | | | ref | | |
| Currently smoking | -1.29 | 0.27 [0.22, 0.34] | < 0.001 | -0.85 | 0.43 [0.11, 1.67] | 0.223 |
| Currently consumed smokeless tobacco | -0.66 | 0.50 [0.43, 0.59] | < 0.001 | -0.16 | 0.85 [0.73, 0.99] | < 0.05 |
| Consumed both smoking and smokeless tobacco | -1.17 | 0.31 [0.23, 0.41] | < 0.001 | -0.45 | 0.64 [0.46, 0.87] | ≤ 0.005 |
| **Alcohol consumption** | | | | | | |
| Never consumed alcohol | ref | | | ref | | |
| Frequently consumed but not a heavy drinker | -0.47 | 0.63 [0.51, 0.77] | < 0.001 | 0.18 | 1.19 [0.93, 1.53] | 0.168 |
| Infrequently consumed but not a heavy drinker | -1.14 | 0.32 [0.26, 0.39] | < 0.001 | 0.13 | 1.14 [0.29, 4.52] | 0.854 |
| Heavy drinker | -1.55 | 0.21 [0.12, 0.38] | < 0.001 | -0.60 | 0.55 [0.28, 1.09] | 0.085 |
| **Region** | | | | | | |

*(Continued)*

**Table 4.** (Continued)

| Variables | b | Model—1 Crude OR (95% CI) | p- value | b | Model—2 Adjusted OR (95% CI) | p- value |
|---|---|---|---|---|---|---|
| North | ref | | | Ref | | |
| West | 0.07 | 1.08 [0.93–1.24] | 0.303 | -0.04 | 0.96 [0.82, 1.13] | 0.621 |
| Northeast | -0.8 | 0.45 [0.37–0.53] | < 0.001 | -0.60 | 0.55 [0.45, 0.68] | < 0.001 |
| East | -0.82 | 0.44 [0.38–0.50] | < 0.001 | -0.63 | 0.53 [0.45, 0.62] | < 0.001 |
| Central | -0.8 | 0.46 [0.39–0.54] | < 0.001 | -0.57 | 0.56 [0.46, 0.67] | < 0.001 |
| South | 0.29 | 1.33 [1.05–1.69] | <0.05 | 0.22 | 1.24 [1.04, 1.48] | < 0.05 |

*OR: Odd Ratio; CI: Confidence Interval; ref: Reference; b: Coefficient; [a] SRH: Self-rated health.

[42,43]. Indians from higher socioeconomic strata consume more calories and fat in their diets and exercise less than those from lower socioeconomic levels, which leads to a higher prevalence of excess weight [44–46]. All of these variables are interconnected, and in the Indian context, upper-caste individuals are often recognised to have better levels of education and economic prosperity than lower-caste individuals. Urban individuals also exhibit comparable traits. These people frequently consume a lot of calories and put forth little effort, which may cause excess weight. The majority of India's lower caste (scheduled tribe) and less wealthy population engages in physical exercise since their economy is based mostly on agriculture. Most of the earlier studies on overweight and obesity in India also portray similar findings [40,47,48].

As previously indicated, this study found that as education levels rise, the likelihood of having excess weight also rises significantly. Many studies support this finding, but few have attempted to determine if education has any beneficial effects on obesity [43]. Siddiqui et al. (2016) [47] found that there is a negative correlation between years of education and the likelihood of having excess weight above a certain threshold level of educational attainment. They found that the likelihood of BMI initially increases with an increasing level of education up to a certain point and then starts declining gradually. This is brought on by a rise in health concerns and awareness among highly educated people [47].

This study further demonstrates that there is a strong correlation between excess weight and chronic diseases, such as hypertension, diabetes, and heart disease. Obesity significantly affects all three ailments, including hypertension [48–53], type 2 diabetes mellitus [48,49,54–56], and heart disease [57–59], all common chronic diseases that are highly costly to our society in terms of health care expenditures and premature morbidity and mortality [60]. It is highly typical for overweight and obese people also to have these chronic conditions.

**Table 5. Health consequences associated with excess weight among older adults (60 years and above) in India using logistic regression models, LASI, 2017–18.**

| Excess weight | Hypertension Adjusted OR (95% CI) | p-value | Diabetes Adjusted OR (95% CI) | p-value | Heart disease Adjusted OR (95% CI) | p-value | Stroke Adjusted OR (95% CI) | p-value | SRH Adjusted OR (95% CI) | p-value |
|---|---|---|---|---|---|---|---|---|---|---|
| No | ref | | ref | | ref | | ref | | ref | |
| Yes | 2.19 [1.92, 2.50] | < 0.001 | 2.14 [1.83, 2.50] | < 0.001 | 1.63 [1.26, 2.11] | < 0.001 | 1.19 [0.89, 1.59] | 0.236 | 0.85 [0.74,0.98] | 0.025 |

* SRH: Self-rated health; All the models are adjusted for the socio-demographic and behavioral factors included in Table 4.

Numerous research has looked at the relationship between obesity and the illnesses mentioned above individually and has come to similar conclusions [61–63].

Physical activity and excess weight have been proven to have a negative correlation in the unadjusted model; those who are constantly active have a lesser risk of having excess weight, and comparable results have been reported in other investigations [64–67]. In this study, smoking behaviour was found to be negatively associated with excess weight. In general, smoking is believed to be a risk factor for weight loss, and many studies on smoking behaviour and body weight revealed that smoking behaviour reduces body weight as smoking is associated with greater energy expenditure, suppressed appetite, and several morbid conditions [68,69].

According to the study, people with excess weight are more likely to report having lower health, and this link holds even after adjusting for the impact of other relevant characteristics such as demographics, socioeconomic position, chronic diseases, and lifestyle choices. The findings of other research across the globe are likewise consistent [70,71]. According to studies, a socioeconomic gradient in health manifests in a way that persons in the lower social strata have worse health [72]; one such illness that disproportionately affects those from lower socioeconomic backgrounds is higher BMI and obesity [73–77]. Given the association between excess weight and SRH, it is possible that other underlying characteristics like socioeconomic status have an impact on self perception of older individuals' health through their influence on excess weight.

A key strength of this research is that we included various socioeconomic and behavioral correlates that play an important role as significant determinants of excess weight and the health consequences of excess weight among older Indians, irrespective of various regions. Another important strength of our study is the inclusion of a nationally representative sample of community-dwelling older adults in India. However, this study has some limitations too. First, the cross-sectional design of the study restricts our ability to infer the causality in the observed associations, and the self-report nature of many of the correlates may lead to reporting biases. Second, several factors such as dietary patterns, food habits, food preferences and food security were not considered in our study. Future research should consider these aspects while analysing the factors associated with excess weight among older adults. Third, BMI measurement in our study does not differentiate between lean or fat mass which can have distinct clinical and biological significance. Fourth, body fat in younger, middle-aged and older adults can have different implications and thus, future studies should focus on age-stratified analysis of factors associated with excess weight. Finally, people from Asian countries have more body fat than people from other regions of the world, and the higher prevalence of excess weight might partially be attributed to the standard cut-off we used. Further investigation is required using the Asian-specific BMI classification and multiple categories of BMI including underweight.

## Conclusion

Findings suggest that female sex, urban place of residence, higher level of education and a higher household economic status were associated with higher prevalence of excess weight among older adults and being diagnosed with hypertension, diabetes and heart disease were the health consequences of excess weight. As such, the difficulties of implementing programs and policies that would lessen the negative consequences of morbidity associated with excess weight among older population must be addressed. The findings further highlight that additional healthy lifestyle practices are needed for the prevention and reduction of excess weight among older adults who have comorbid conditions, such as diabetes and hypertension. In order to improve older adults' functional status and prevent them from becoming disabled

and consequently experiencing poor quality of life, policymakers and healthcare professionals must consider interventions addressing excess weight while developing disease-specific management programmes.

## Author Contributions

**Conceptualization:** Amiya Saha, T. Muhammad.

**Data curation:** Amiya Saha.

**Formal analysis:** Amiya Saha.

**Methodology:** Amiya Saha, Bittu Mandal, Papai Barman.

**Software:** Amiya Saha.

**Supervision:** Amiya Saha, Bittu Mandal, Mihir Adhikary, Papai Barman.

**Visualization:** Amiya Saha.

**Writing – original draft:** Amiya Saha, T. Muhammad, Bittu Mandal, Mihir Adhikary, Papai Barman.

**Writing – review & editing:** Amiya Saha, T. Muhammad.

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
