## [Decision Letter · Decision Letter 0]

18 May 2023

PONE-D-23-04836Determinants of obesity in relation to socioeconomic and health status among older adults in India: Evidence from a cross-sectional survey, 2017-18PLOS ONE

Dear Dr. Saha,

Thank you for submitting your manuscript to PLOS ONE. After careful consideration, we feel that it has merit but does not fully meet PLOS ONE’s publication criteria as it currently stands. Therefore, we invite you to submit a revised version of the manuscript that addresses the points raised during the review process.

We look forward to receiving your revised manuscript.

Kind regards,

Chandan Kumar, Ph.D.

Academic Editor

PLOS ONE

Journal Requirements:

“The National Institute on Aging (R01 AG042778, R01 AG030153), the United Nations Population Fund, and the Government of India's Ministry of Health and Family Welfare all provided funding for the Longitudinal Aging Study in India Project.”

3. We note that Figure 2 in your submission contain [map/satellite] images which may be copyrighted. All PLOS content is published under the Creative Commons Attribution License (CC BY 4.0), which means that the manuscript, images, and Supporting Information files will be freely available online, and any third party is permitted to access, download, copy, distribute, and use these materials in any way, even commercially, with proper attribution. For these reasons, we cannot publish previously copyrighted maps or satellite images created using proprietary data, such as Google software (Google Maps, Street View, and Earth). For more information, see our copyright guidelines: http://journals.plos.org/plosone/s/licenses-and-copyright.

Reviewers' comments:

Reviewer's Responses to Questions

**Comments to the Author**

1. Is the manuscript technically sound, and do the data support the conclusions?

Reviewer #1: Partly

Reviewer #2: Partly

2. Has the statistical analysis been performed appropriately and rigorously? 

Reviewer #1: Yes

Reviewer #2: Yes

3. Have the authors made all data underlying the findings in their manuscript fully available?

Reviewer #1: Yes

Reviewer #2: Yes

4. Is the manuscript presented in an intelligible fashion and written in standard English?

Reviewer #1: Yes

Reviewer #2: No

5. Review Comments to the Author

Reviewer #1: 1. Author should include some important interaction findings in the abstracts’ results section.

2. Is the LASI data available in any online repository? A link can be included.

3. Please do not mention “untouchables”: SC and ST are specifically defined in the Indian Constitution. A reference can be added.

4. The discussion should begin with the hypothesis behind the study.

5. Line 337: Please include a reference for this statement about Numerous research has looked at the relationship between obesity and the illnesses mentioned above individually and has come to similar conclusions.

6. Percentage mentioned in the should be in round figure as it added in the results.

7. Why authors have given crude odds for factors as already given bivariate association in previous table.

8. Figure is not clearly visible. Authors may add new figure with high resolution.

9. What is unique about this study? The association between obesity and NCD is well known. Even obesity leads to diseases like hypertension, diabetes, CVD, and stroke. In this study, result showed a higher percentage of older adults with hypertension (30.3%), diabetes (46.3%), and CVD (41.3%) were obese. Previous research also have similar findings like person with higher socio-economic have higher likelihood of obesity.

Reviewer #2: An interesting study as its used national epidemiological data, however there is lack of focus in the contents, unclear variables definitions and discussion. Moderate alcohol intake is part of the variables in the objectives but no proper definition and description in the results/discussion. Some of the results mention in the discussion and conclusion are not from the analysis presented. There are some ambiguous descriptions which need revision. Suggest to do proper language editing prior to final submission.

Abstract - In the methodology, I would suggest to include the BMI level definition for obesity used in this study. In the conclusion, the statement 'The present study suggests that introducing preventative measures such...' In my opinion, the study did not assess such preventative measures, hence it is not fair to claim such measures particularly food policies and healthy diet were not part of the variables studied.

Line 94 - please revise the statement for reference 18, they used BMI equal or more than 25 to describe excess weight (overweight and obese), not obesity per se.

Line 129-135: Objectives - Please revise to make them consistent with results and discussion (My suggestion: focus on the three areas - sociodemographic, health profiles and health behaviours).

a) what does comprehensive prevalence means?

b) what do you mean by '...independent prediction'? Age is not included in the results, but mention in this objective. Education is mentioned in b) and d). Why 'education' specifically chosen as one of the focus?

c) Moderate alcohol - not described in variable definition and not clear in the results/ discussion.

Study Population

Line 159: What does '...(including Sikkim)' means?

Variable definitions - this section needs further descriptions in the text, please elaborate the operational definitions used.

Line 142 - Please use own words instead of '....' from reference 35.

Line 168 - the number stated is the number of participants included in this analysis, not sample size determination.

Line 171: Ethic - please provide ethics approval statement for the LASI study. Modify last sentence in line 176.

Line 172 - typo 'publicly' not 'publically'

1. Line 182-186: The definition provided is the standard definition for overweight and obesity, not Asian- specific definition. Please clarify, which definition is being used for this manuscript.

2. Sociodemographic data:

- how does rural and urban being defined in the study

- Caste: please explain what does ST, SC, OBC means and how are they being identified (which one is lower or higher caste?)

-Wealth: how does the wealth being defined (e.g. how do you classify poorest - any specific income level?)

3. Disease profile:

- Hypertension: was it self reported or any blood pressure measurement done for the study population (e.g. how do participants know about mean systolic or diastolic blood pressure?)

- What does cardiovascular diseases consist of? How does participants know they are having cardiovascular disease? Some definition includes hypertension and stroke as cardiovascular disease, how about this study?

(Table 1, line 189)

4. Health behaviours:

- How does physical activity being assessed? What does it consist of?

- Tobacco consumption: How is it being classified? (Confusing - third group 'smoke tobacco but do not smoking', what does it mean?

- Alcohol consumption: How is it being classified? Which group describes 'moderate consumption'?

-How does self rated health being assessed, was it only two options of 'good' and 'poor' were given to participants

Statistical Analysis:

Not clear the importance of stating the equation for logistic regression.

There is a need for clear description the meaning of Model 1 and model 2 for regression, how do you select which variables for multiple logistic regression.

Results:

Line 227 and Table 3, please be consistent with the topic, why does it suddenly change to noncommunicable diseases.

Line 255 onwards - suggest to focus, simplify and describe the important findings from the results. What does the results in Table 4 means? Instead of repetitively writing back the findings in the Table 4 in the text.

Please be careful with the interpretation of the regression analysis, few statements of 'more obese' seem inappropriate (line 258, 260, 268). MPCE quintile - I am not sure what does this means.

Suggest to focus discussed the results with adjusted OR, instead of both. Line 275 - wrong result for SRH.

Suggest to focus to the study objectives.

Discussion - please highlight the areas related to the study objectives

Limitation - Confusing, particularly second, third and fourth

Conclusion is confusing, not consistent with the study objectives - suggest to revise, focus to the study findings and clearly state recommendations. Nutrition is not being assessed but included in the conclusion as part of findings?

line 382 - 'age' was not reported in the study result/ analysis, but described as a factor for obesity.

Reference 35 - please follow proper suggested citation:

International Institute for Population Sciences (IIPS), National Programme for

Health Care of Elderly (NPHCE), MoHFW, Harvard T. H. Chan School of

Public Health (HSPH) and the University of Southern California (USC) 2020.

Longitudinal Ageing Study in India (LASI) Wave 1, 2017-18, India Report,

International Institute for Population Sciences, Mumbai.

6. PLOS authors have the option to publish the peer review history of their article (what does this mean?). If published, this will include your full peer review and any attached files.

Reviewer #1: **Yes: **Dr. Pradeep Kumar

Reviewer #2: No

---

## [Author Response · Author response to Decision Letter 0]

31 May 2023

Comments to the Author

1. Is the manuscript technically sound, and do the data support the conclusions?

Reviewer #1: Partly

Reviewer #2: Partly

2. Has the statistical analysis been performed appropriately and rigorously?

Reviewer #1: Yes

Reviewer #2: Yes

3. Have the authors made all data underlying the findings in their manuscript fully available?

Reviewer #1: Yes

Reviewer #2: Yes

4. Is the manuscript presented in an intelligible fashion and written in standard English?

Reviewer #1: Yes

Reviewer #2: No

5. Review Comments to the Author

Reviewer #1: 

1. Author should include some important interaction findings in the abstracts’ results section.

Thank you for the comment. We have reported the important findings from the multivariable analysis in the abstract’s results.

2. Is the LASI data available in any online repository? A link can be included.

Yes, the LASI data are available in the repository of the Gateway to Global Aging Data and the accessible link has been provided now.

3. Please do not mention “untouchables”: SC and ST are specifically defined in the Indian Constitution. A reference can be added.

Done! The variable description table now includes a description of caste categories in our study. Relevant citations are provided as well.

4. The discussion should begin with the hypothesis behind the study.

Thank you for this comment. The beginning of the discussion is now with the objectives of our study.

5. Line 337: Please include a reference for this statement about “Numerous research has looked at the relationship between obesity and the illnesses mentioned above individually and has come to similar conclusions.”

We appreciate your suggestion. Relevant citations are provided now.

6. Percentage mentioned in the should be in round figure as it added in the results.

Thank you for the suggestion. The percentages reported in the results are now in round figures.

7. Why authors have given crude odds for factors as already given bivariate association in previous table.

Sure thing! We have now removed the paragraph that reports the unadjusted results from the multivariable analysis in the results section.

8. Figure is not clearly visible. Authors may add new figure with high resolution.

Thank you for the suggestion! In the revised manuscript, we have provided the figures with improved quality. Please let us know if this helps.

9. What is unique about this study? The association between obesity and NCD is well known. Even obesity leads to diseases like hypertension, diabetes, CVD, and stroke. In this study, result showed a higher percentage of older adults with hypertension (30.3%), diabetes (46.3%), and CVD (41.3%) were obese. Previous research also have similar findings like person with higher socio-economic have higher likelihood of obesity.

We appreciate your comment. We also agree that the readers of the journal will look for the novelty in our study. To that end, we have revised all the sections of the manuscript and now we focus on the socioeconomic, behavioural and health-related correlates of excess weight among older Indians. There is dearth of such studies conducted among community-dwelling older Indians. Especially, the large sample size in the LASI data was a major strength of our study. These are highlighted now.

Reviewer #2: 

An interesting study as its used national epidemiological data, however there is lack of focus in the contents, unclear variables definitions and discussion. Moderate alcohol intake is part of the variables in the objectives but no proper definition and description in the results/discussion. Some of the results mention in the discussion and conclusion are not from the analysis presented. There are some ambiguous descriptions which need revision. Suggest to do proper language editing prior to final submission.

We apologize for the errors. This was an oversight on our part. In the revised manuscript, we have provided the proper definitions for the selected variables including the alcohol intake. We have revised the discussion and conclusions based on our findings. We have also corrected for typos, grammar, punctuation, and sentence structuring errors.

Abstract - In the methodology, I would suggest to include the BMI level definition for obesity used in this study. In the conclusion, the statement 'The present study suggests that introducing preventative measures such...' In my opinion, the study did not assess such preventative measures, hence it is not fair to claim such measures particularly food policies and healthy diet were not part of the variables studied.

We appreciate your comment. The conclusion section is revised to connect with the current findings.

Line 94 - please revise the statement for reference 18, they used BMI equal or more than 25 to describe excess weight (overweight and obese), not obesity per se.

Thank you! The statement is modified now.

Line 129-135: Objectives - Please revise to make them consistent with results and discussion (My suggestion: focus on the three areas - sociodemographic, health profiles and health behaviours).

a) what does comprehensive prevalence means?

This was an oversight on our part, we apologise. We have removed the term “comprehensive prevalence” in the revised manuscript.

b) what do you mean by '...independent prediction'? Age is not included in the results, but mention in this objective. Education is mentioned in b) and d). Why 'education' specifically chosen as one of the focus?

Thank you for this helpful suggestion. The statements on objectives of the study are revised and corrected for these errors.

c) Moderate alcohol - not described in variable definition and not clear in the results/ discussion.

Done! The description of the measurements including that of alcohol and tobacco consumption has been provided in the revised manuscript.

Study Population

Line 159: What does '...(including Sikkim)' means?

We apologise for this error. These words (including Sikkim) are removed now in the revised manuscript.

Variable definitions - this section needs further descriptions in the text, please elaborate the operational definitions used.

Elaborate definitions of the selected variables have been added now.

Line 142 - Please use own words instead of '....' from reference 35.

We appreciate this comment. We have revised the data section and paraphrased the statements.

Line 168 - the number stated is the number of participants included in this analysis, not sample size determination.

Thanks for your comment. The statement is revised to offer more clarity.

Line 171: Ethic - please provide ethics approval statement for the LASI study. Modify last sentence in line 176.

Ethics approval statement is provided now. And the line 176 is modified to offer more clarity.

Line 172 - typo 'publicly' not 'publically'

We have corrected this typo. We appreciate your comment.

1. Line 182-186: The definition provided is the standard definition for overweight and obesity, not Asian- specific definition. Please clarify, which definition is being used for this manuscript.

The BMI categories were based on the standard definition and according to the classification of the World Health Organization. This is clarified now in the revised manuscript. 

2. Sociodemographic data:

- how does rural and urban being defined in the study

We have clarified this in the table of variable description in the revised manuscript. We appreciate your suggestion.

- Caste: please explain what does ST, SC, OBC means and how are they being identified (which one is lower or higher caste?)

We appreciate your comment on caste. We have now clearly stated of caste stratification in the revised manuscript. 

-Wealth: how does the wealth being defined (e.g. how do you classify poorest - any specific income level?)

We appreciate your comment on wealth classification. We have now clarified the wealth classification in the revised manuscript. 

3. Disease profile:

- Hypertension: was it self reported or any blood pressure measurement done for the study population (e.g. how do participants know about mean systolic or diastolic blood pressure?)

We have clarified this in the table of variable description in the revised manuscript. We appreciate your suggestion.

- What does cardiovascular diseases consist of? How does participants know they are having cardiovascular disease? Some definition includes hypertension and stroke as cardiovascular disease, how about this study?

Thank you for the comment! The term “cardiovascular diseases” in the earlier submission included heart diseases only. To offer more clarity, the term is replaced with “heart disease” in the revised manuscript throughout.

(Table 1, line 189)

4. Health behaviours:

- How does physical activity being assessed? What does it consist of?

We have clarified this in the table of variable description in the revised manuscript. We appreciate your suggestion.

- Tobacco consumption: How is it being classified? (Confusing - third group 'smoke tobacco but do not smoking', what does it mean?

We have clarified this in the table of variable description in the revised manuscript. We appreciate your suggestion.

- Alcohol consumption: How is it being classified? Which group describes 'moderate consumption'?

We have clarified this in the table of variable description in the revised manuscript. We appreciate your suggestion.

-How does self rated health being assessed, was it only two options of 'good' and 'poor' were given to participants

We have clarified this in the table of variable description in the revised manuscript. We appreciate your suggestion.

Statistical Analysis:

Not clear the importance of stating the equation for logistic regression.

Thank you for the comment. The equation for logistic regression is removed now.

There is a need for clear description the meaning of Model 1 and model 2 for regression, how do you select which variables for multiple logistic regression.

We appreciate your comment. As such, in the statistical analysis section, we have provided more description of models and the control variables.

Results:

Line 227 and Table 3, please be consistent with the topic, why does it suddenly change to non-communicable diseases.

This was an oversight on our part, we apologize. The titles of the tables and the description have been revised and made consistent throughout. We report socio-economic, behavioural and health-related factors associated with obesity, as mentioned in the title of our study.

Line 255 onwards - suggest to focus, simplify and describe the important findings from the results. What does the results in Table 4 means? Instead of repetitively writing back the findings in the Table 4 in the text.

Sure thing! The results section has been revised and shortened as per suggestion.

Please be careful with the interpretation of the regression analysis, few statements of 'more obese' seem inappropriate (line 258, 260, 268). MPCE quintile - I am not sure what does this means.

Thank you for the comment. The results section is revised extensively and corrected for these errors. A description of monthly per capita consumption expenditure (MPCE) quintile has been provided in the methods section.

Suggest to focus discussed the results with adjusted OR, instead of both. Line 275 - wrong result for SRH.

Suggest to focus to the study objectives.

We appreciate your suggestions, thank you. The reported results with unadjusted ORs are removed now and the results section now focuses on the adjusted estimates. The result with adjusted OR of SRH is also provided now.

Discussion - please highlight the areas related to the study objectives

The discussion is revised and it now focuses on the main findings of our study.

Limitation - Confusing, particularly second, third and fourth

Thank you! The limitations have been revised in the updated version with detailed explanation.

Conclusion is confusing, not consistent with the study objectives - suggest to revise, focus to the study findings and clearly state recommendations. Nutrition is not being assessed but included in the conclusion as part of findings?

We appreciate your suggestion. The conclusion part is significantly revised to focus on the study findings and provide clear implications.

line 382 - 'age' was not reported in the study result/ analysis, but described as a factor for obesity.

Thank you for pointing this out! The conclusion is revised accordingly and removed mentioning age, and the need for age-stratified analysis has been mentioned in the limitations.

Reference 35 - please follow proper suggested citation:

International Institute for Population Sciences (IIPS), National Programme for

Health Care of Elderly (NPHCE), MoHFW, Harvard T. H. Chan School of

Public Health (HSPH) and the University of Southern California (USC) 2020.

Longitudinal Ageing Study in India (LASI) Wave 1, 2017-18, India Report,

International Institute for Population Sciences, Mumbai.

Thank you for taking your time to review our manuscript. The references have been updated now. 

We are grateful for your comments and suggestions.

---

## [Decision Letter · Decision Letter 1]

16 Aug 2023

PONE-D-23-04836R1Socio-demographic, behavioral and health-related correlates of excess weight among older adults in India: Evidence from a cross-sectional study, 2017-18PLOS ONE

Dear Dr. Saha,

Thank you for submitting your manuscript to PLOS ONE. After careful consideration, we feel that it has merit but does not fully meet PLOS ONE’s publication criteria as it currently stands. Therefore, we invite you to submit a revised version of the manuscript that addresses the points raised during the review process.The revised manuscript still needs thorough language editing in some of its sections. Please refer to the attached edited copy of the manuscript (in track-change mode), I’ve tried to edit wherever possible, but they are not exhaustive. Please consider the editing done in the attached copy of the manuscript, and accordingly revise the manuscript.Authors are advised to reconsider the variables included in the health-related factors. Hypertension, Diabetes, stoke, and heart diseases are not the determinants of obesity, but this is the other way around. These health issues are certainly associated with obesity, but they cannot be considered as its predictors. Authors can present these health issues as a consequence of overweight/obesity.LASI data does not provide a wealth quintile; it is an MPCE quintile, and they can’t be used interchangeably. So, replace the wealth quintile with the MPCE quintile, wherever used.Please consider other comments and suggestions mentioned in the attached edited copy of the manuscript.

We look forward to receiving your revised manuscript.

Kind regards,

Chandan Kumar, Ph.D.

Academic Editor

PLOS ONE

Journal Requirements:

Reviewers' comments:

Reviewer's Responses to Questions

**Comments to the Author**

1. If the authors have adequately addressed your comments raised in a previous round of review and you feel that this manuscript is now acceptable for publication, you may indicate that here to bypass the “Comments to the Author” section, enter your conflict of interest statement in the “Confidential to Editor” section, and submit your "Accept" recommendation.

Reviewer #2: (No Response)

2. Is the manuscript technically sound, and do the data support the conclusions?

Reviewer #2: Yes

3. Has the statistical analysis been performed appropriately and rigorously? 

Reviewer #2: Yes

4. Have the authors made all data underlying the findings in their manuscript fully available?

Reviewer #2: Yes

5. Is the manuscript presented in an intelligible fashion and written in standard English?

Reviewer #2: Yes

6. Review Comments to the Author

Reviewer #2: Thank you and well done for your much clearer revised manuscript.

Abstract:

Method

Line 39 - suggest to include definition of BMI 25 and above as excess weight.

Table 1:

- a few typos

Description in Consumption of tobacco - line 1 'redondents', line 5 'repond'

Description in Consumption of alcohol - line 2 'redondents'

Description Self-rated health - line 4 '...coded as bad' ( I think it should be '..coded as poor')

Table 2 & 3 - please be consistent with decimal points.

7. PLOS authors have the option to publish the peer review history of their article (what does this mean?). If published, this will include your full peer review and any attached files.

Reviewer #2: No

---

## [Author Response · Author response to Decision Letter 1]

23 Aug 2023

Response to the Editor comments

Thank you for the thorough review of our paper! We have made the changes as per the Editor’s comments and used track changes to highlight the revisions within the manuscript. We provide an explanation below on how we address each comment provided in the edited copy of the manuscript. Kindly let us know if further changes are needed.

[p 2, line 41- 43]

The highest prevalence can be in one state, not in so many states. What is the rationale for choosing 33.2% prevalence cut-off? Authors can list out states above the national average. 

The higher levels of excess weight (than the national average of 22.7%) were observed among older adults in states like Kerala, Karnataka, Himachal Pradesh, Punjab, and Sikkim………...has been updated in the revised manuscript.

[p 2, line 44- 45]

Replace respondents with older adults throughout the manuscripts, wherever the authors are presenting estimates. The estimates are presented for the study population, i.e., older adults.

Done! The word “older adults” is used throughout in the revised manuscript.

[p 2, line 47- 50]

Do the authors really want to highlight this association so explicitly? Education is correlated with the economic status of the population; this may not be the key factor in making someone overweight. Please consider the line of argument the authors want to put through their analysis and interpretation.

The statements are modified accordingly. They now read as follow:

Higher level of education is significantly positively correlated with excess weight. Similarly, higher household wealth index was significantly positively correlated with excess weight [OR: 1.77, CI: 1.46, 2.15].

[p 8, line 73- 75]

If the source of the information for these two sentences is the same, they can also be clubbed together, otherwise, the reference must be provided to the first sentence.

The statements are now clubbed together and the citation is provided at the end, and it reads as follows:

About 104 million people in India are 60 or older, constituting 8.6% of the total population, and by 2050, the percentage is expected to rise to 20% of the population.

[p 8, line 175- 177]

Authors are advised to reconsider the variables included in the health-related factors. Hypertension, Diabetes, stoke, and Heart diseases are not the determinants of obesity, but this is the other way around. These health issues are certainly associated with obesity, but they cannot be considered as its predictors. Authors can present these health issues as a consequence of obesity… 

Done! The health-related variables are now removed from the predictors and a separate table is provided now to report the health-related consequences of excess weight among older adults.

[p 10, Table 1]

This should be replaced with MPCE Quintile

Done! The word “Wealth quintiles” is replaced by “MPCE quintiles” now.

[p 15, Table 3]

Combine the lower and upper bound of the 95% CI to present them in parentheses, e.g., (16.3, 18.9).

Present the numbers with single digit after decimal.

Thank you! The table is revised to incorporate both the suggestions.

[p 16, Table 3]

Also, provide the estimates for India (Total) in the last row.

Done!

[p 21, line 305-308]

This is the point the Authors have to recognize that these health effects are the resultant factor of overweight/obesity, not the determinants. So, present these associations in a separate section/table.

Done! In the revised manuscript, a separate table is provided now to report the health-related consequences of excess weight among older adults.

Response to the Reviewer comments

Reviewer #2: Thank you and well done for your much clearer revised manuscript.

Thank you for the thorough review of our paper! We have made the changes as per the Reviewer’s comments and used track changes to highlight the revisions within the manuscript. Kindly let us know if further changes are needed.

Method

Line 39 - suggest to include definition of BMI 25 and above as excess weight.

[p – 37 -38]

Done! Now the definition has been provided with much more clarity.

Table 1:

- a few typos

Description in Consumption of tobacco - line 1 'redondents', line 5 'repond'

Done!

Description in Consumption of alcohol - line 2 'redondents'

Done!

Description Self-rated health - line 4 '...coded as bad' ( I think it should be '..coded as poor')

Done!

Table 2 & 3 - please be consistent with decimal points.

Done!

[p 13, 15]

---

## [Editor Report · Decision Letter 2]

30 Aug 2023

PONE-D-23-04836R2Socio-demographic and behavioral correlates of excess weight and its health consequences among older adults in India: Evidence from a cross-sectional study, 2017-18PLOS ONE

Dear Dr. Saha,

Thank you for submitting your revised manuscript, but I can see some modifications in the manuscript still needs to be done. Please refer to the following comment, which is also mentioned in the attached copy of the manuscript. 

**"The interpretation of Table 5 needs to be modified. The odds ratios of the logistic regression models suggest that older adults with excess weight were 2.19 times more likely to be hypertensive.... (similar interpretation in case of other diseases too), not the other way around as interpreted by the authors."** **Authors are also requested to go through the entire manuscript again to ignore such mistakes, especially where the "health-related factors" were mentioned in the earlier manuscript and highlighted by the Editor. The health-related factors must be replaced with health consequences of excess weight wherever mentioned in the manuscript, and interpreted and discussed in similar lines. ** We invite you to submit a revised version of the manuscript that addresses the points raised here. 

We look forward to receiving your revised manuscript.

Kind regards,

Chandan Kumar, Ph.D.

Academic Editor

PLOS ONE
---

## [Author Response · Author response to Decision Letter 2]

7 Sep 2023

Thank you for the thorough review of our paper! We have made the changes as per the Editor’s comments and used track changes to highlight the revisions within the manuscript. We address each comment in the revised manuscript. Kindly let us know if further changes are needed.

"The interpretation of Table 5 needs to be modified. The odds ratios of the logistic regression models suggest that older adults with excess weight were 2.19 times more likely to be hypertensive.... (similar interpretation in case of other diseases too), not the other way around as interpreted by the authors."

The interpretation of Table 5 is modified accordingly. They now read as follow:

“The odds ratios of the logistic regression models (Table 5) suggest that older adults with excess weight were 2.19 times [OR: 2.19, 95% CI: 1.92, 2.50], 2.14 times [OR: 2.14, 95% CI: 1.83, 2.50] and 1.63 times [OR: 1.63, 95% CI: 1.26, 2.11] significantly more likely to be hypertensive, diabetic and have heart disease. Similarly, older adults with excess weight were 0.85 times [OR: 0.85, 95% CI: 0.74, 0.98] significantly less likely to report good SRH in this study.”

Authors are also requested to go through the entire manuscript again to ignore such mistakes, especially where the "health-related factors" were mentioned in the earlier manuscript and highlighted by the Editor. The health-related factors must be replaced with health consequences of excess weight wherever mentioned in the manuscript, and interpreted and discussed in similar lines. 

Done!

We have gone through the whole manuscript and replaced the above-mentioned word in the entire manuscript!

Thank you for your time and support!

---

## [Editor Report · Decision Letter 3]

11 Sep 2023

Socio-demographic and behavioral correlates of excess weight and its health consequences among older adults in India: Evidence from a cross-sectional study, 2017-18

PONE-D-23-04836R3

Dear Dr. Saha,

We’re pleased to inform you that your manuscript has been judged scientifically suitable for publication and will be formally accepted for publication once it meets all outstanding technical requirements.

Kind regards,

Chandan Kumar, Ph.D.

Academic Editor

PLOS ONE

---

## [Editor Report · Acceptance letter]

26 Sep 2023

PONE-D-23-04836R3 

Socio-demographic and behavioral correlates of excess weight and its health consequences among older adults in India: Evidence from a cross-sectional study, 2017-18 

Dear Dr. Saha:

I'm pleased to inform you that your manuscript has been deemed suitable for publication in PLOS ONE. Congratulations! Your manuscript is now with our production department. 

Kind regards, 

on behalf of

Dr. Chandan Kumar 

Academic Editor

PLOS ONE